# Breaking Dormancy and Increasing Restoration Success of Native Penstemon Species Using Gibberellic Acid Seed Coatings and U-Shaped Furrows

**DOI:** 10.3390/plants12234005

**Published:** 2023-11-28

**Authors:** Amber J. Johnson, Bradley Geary, April Hulet, Matthew D. Madsen

**Affiliations:** Department of Plant & Wildlife Sciences, Brigham Young University, Provo, UT 84602, USA; brad_geary@byu.edu (B.G.); april_hulet@byu.edu (A.H.); matthew.madsen@byu.edu (M.D.M.)

**Keywords:** dormancy, planting season, forbs, microsite, habitat

## Abstract

Many plant species exhibit strong seed dormancy. This attribute benefits the species’ long-term survival but can impede restoration when rapid establishment is required. Soaking seeds in gibberellic acid (GA_3_) can overcome dormancy and increase germination but this treatment may not be effective outside the laboratory. An easier and potentially more effective method to apply this hormone is to coat seeds with a GA_3_-impregnated polymer. Seed dormancy can also be mitigated by creating a favorable microsite with increased soil moisture. We compared the emergence and establishment of penstemon seeds coated with GA_3_ to those of uncoated seeds planted in shallow drill rows versus deep, U-shaped furrows. Overall, 6 times more Palmer’s penstemon (*Penstemon palmeri*; *p* < 0.01) and 21 times more thickleaf penstemon (*P. pachyphyllus*; *p* < 0.001) established when coated with GA_3_, but GA_3_ coating did not affect the establishment of firecracker penstemon (*P. eatonii*; *p* = 1). Establishment was higher from deep furrows than shallow rows (*p* < 0.001). These results indicate that GA_3_ seed coating and deep, U-shaped furrows may improve the restoration success of some native forbs by breaking dormancy and providing a favorable microsite. Land managers could use these techniques to restore native forbs in dry, disturbed areas.

## 1. Introduction

While habitat restoration efforts have historically focused on perennial grasses, native forbs are also important to reestablish in ecosystems after a disturbance [1,2,3]. Forbs have value for wildlife and livestock forage, pollinator use, and ecosystem resilience [4,5,6]. The use of these species has historically been limited due to high costs and low success in forcing plants to germinate and emerge [1,3,7]. One reason for the low emergence of these forbs is that many of these species exhibit strong physiological dormancy [8,9,10]. Seed dormancy is especially prevalent in arid environments, where this characteristic minimizes risk by postponing germination to avoid conditions where the seedling is unlikely to survive, such as freezing temperatures and drought [10,11,12]. For restoration efforts, however, this adaptation can be a limitation. After a disturbance such as fire, it is often critical to establish plants quickly to prevent erosion and invasion of exotic weeds [3,13]. Therefore, to improve the establishment of dormant forb species, it may be advantageous to break seed dormancy so the plant can rapidly establish and not remain dormant for several seasons.

Previous studies on overcoming dormancy have shown that gibberellic acid (GA_3_) increased germination and emergence of various native forb species that are physiologically dormant [14,15,16]. These studies applied the hormone by soaking seeds in a solution of GA_3_, ethanol, and water [14,15,16]. Soaking seeds in GA_3_ can overcome dormancy and increase germination, but this treatment may not be effective outside of the laboratory [3]. An easier and potentially more effective method to apply this hormone is to coat seeds with a GA_3_-impregnated polymer [17]. This coating technique slowly releases the active ingredient to the seed as the coating breaks down in the soil. This slow release may extend the effects of the gibberellic acid to the seeds as they germinate and as the seedlings grow.

Using GA_3_ to decrease the time to germination may make planting dormant species in spring possible. Generally, cold, wet conditions over winter are needed to break the physiological dormancy of many forbs [11,18,19]. By using GA_3_ to decrease the time to germination, spring planting may be an alternative to traditional fall restoration plantings, which may improve the survival of seedlings [20]. When seeds germinate in the fall or winter, they are exposed to pathogens and freezing conditions for months before emerging in the spring [21,22,23]. This can result in a severe bottleneck between germination and emergence [23,24,25]. Planting GA_3_-coated seeds in the spring may circumvent the worst of that bottleneck by allowing seeds to germinate when conditions are prime for emergence.

Another technique to improve the emergence and establishment of dormant native species is to use furrows to create favorable microsites [26,27,28]. Within deep (~15 cm) furrows, soil moisture is higher and temperatures are more moderate than they are in the surrounding areas [27]. Deep furrows can be especially useful in arid landscapes like the Great Basin region of the United States, where seedings commonly fail due to drought conditions [2,22]. The shape of the furrow can also be modified to enhance the microsite the seed is planted within. Traditional furrows are often created in a V-shape [27,28]. Small seeds planted in V-shaped furrows, however, are at risk of being buried too deep from soil sloughing into the furrow [28,29]. Ideally, seeds are planted in the soil at a depth of two to three times the width of the seed [7,18]. Since many forbs in the Great Basin have very small seeds, and therefore need to be seeded only a few millimeters below the soil surface, they can be at risk of being sown too deep [29,30,31]. U-shaped furrows with a wide bottom may decrease the amount of soil that collapses onto the seeds after planting by increasing the distance between the seed and the sidewall of the furrow [28]. Thus, when the sidewall of the furrow erodes into the furrow, it is less likely to reach the location of the seed. Subsequently, U-shaped furrows may provide the same microsite benefits of traditional V-shaped furrows without risking small forb seeds being buried too deep [27,28].

A large genus of small-seeded forbs common in the Great Basin are penstemons (*Penstemon* Mitch.) [11,32]. Several species of penstemon are commonly used in restoration efforts in western North America as they are valuable for wildlife, livestock, and pollinators [11,33,34,35]. Additionally, these species can assist in erosion control and protect soils [33,34]. Many penstemon species show strong physiological dormancy [9,11] and respond positively to applications of GA_3_ [11,36]. We selected three penstemon species to test our technologies: Palmer’s penstemon (*Penstemon palmeri* A. Gray), thickleaf penstemon (*P. pachyphyllus* A. Gray ex Rydb.), and firecracker penstemon (*P. eatonii* A. Gray). Preliminary laboratory research has shown that GA_3_ coatings may decrease dormancy and increase the emergence of these species [17].

In this study, we tested GA_3_ seed coating, fall and spring planting, and U-shaped furrows to improve the restoration success of penstemon species at degraded rangeland sites across Utah, USA. We compared the emergence and establishment of native penstemon seeds coated with GA_3_ to those of uncoated seeds sown in late fall and early spring. Additionally, we compared GA_3_-coated and uncoated seeds sown in shallow, drill rows and deep, U-shaped furrows. We predicted that seeds coated with GA_3_ would emerge and establish at a higher rate than uncoated seeds. We also predicted that GA_3_-coated seeds would emerge and establish at a higher rate when sown in the spring than in the fall. Further, we predicted there would be higher emergence and establishment in deep, U-shaped furrows than in shallow, drill rows.

## 2. Results

### 2.1. Palmer’s Penstemon

Fall planting, seed coating, and deep furrows all improved seedling emergence, with the combination of these treatments resulting in the highest number of Palmer’s penstemon seedlings (Figure 1). Seed treatment had the strongest influence on seedling emergence (*F* = 13.8, *p* < 0.001), followed by furrows (*F* = 7.5, *p* = 0.006), and planting season (*F* = 6, *p* = 0.01). There were no significant interactions, so the model was analyzed without interactions. When planted in the fall, no Palmer’s penstemon emerged in shallow rows unless the seed was coated with GA_3_ (*p* < 0.001; Figure 1). Fall planting in deep furrows had five-fold higher emergence from seeds coated with GA_3_ ($\bar{x}$ = 1.25 plants m^−1^) than from uncoated seeds ($\bar{x}$ = 0.25 plants m^−1^; *p* = 0.005). Overall, two times more seedlings emerged when planted in the fall ($\bar{x}$ = 0.52 plants m^−1^) than when planted in the spring ($\bar{x}$ = 0.21 plants m^−1^; *p* = 0.01). All treatments had relatively minimal emergence for spring planting, except for GA_3_-coated seeds planted in deep furrows. Spring planting in deep furrows had seven times higher emergence from seeds coated with GA_3_ ($\bar{x}$ = 0.58 plants m^−1^) than from uncoated seeds ($\bar{x}$ = 0.08 plants m^−1^; *p* < 0.005; Figure 1). Across both planting seasons and seed treatments, three times more Palmer’s penstemon emerged when planted in deep furrows ($\bar{x}$ = 0.54 plants m^−1^) compared to those in shallow rows ($\bar{x}$ = 0.19 plants m^−1^; *p* = 0.006).

As with seedling emergence, the density of established Palmer’s penstemon at the end of the growing season was highest for GA_3_-coated seeds planted in the fall in deep furrows (Figure 2). There was a significant interaction between seed treatment and planting season (*p* = 0.007), so we included that interaction in the model. When planted in the fall in shallow rows, there was eight times higher establishment from GA_3_-coated seeds ($\bar{x}$ = 0.67 plants m^−1^) than uncoated seeds ($\bar{x}$ = 0.08 plants m^−1^; *p* = 0.005). This increase in establishment was more pronounced when GA_3_-coated seeds were sown in deep furrows. For fall planting, the establishment of GA_3_-coated seeds planted in deep furrows ($\bar{x}$ = 1.67 plants m^−1^) was 20 and 10 times higher than that of uncoated seeds planted in shallow rows ($\bar{x}$ = 0.08 plants m^−1^; *p* < 0.001) and deep furrows ($\bar{x}$ = 0.17 plants m^−1^; *p* = 0.006), respectively (Figure 2). For spring planting, plant establishment did not vary by seed treatment or furrow type (*p* = 1; Figure 2).

### 2.2. Thickleaf Penstemon

As with Palmer’s penstemon, fall planting, seed coating, and deep furrows all improved seedling emergence of thickleaf penstemon, with the combination of these treatments resulting in the highest number of plants (Figure 1). Seedling emergence was most influenced by seed treatment (*F* = 17.8, *p* < 0.001) and planting season (*F* = 17.8, *p* < 0.001) but was also influenced by furrows (*F* = 4.8, *p* = 0.03). Thickleaf penstemon had relatively minimal emergence across all planting season and depth combinations unless coated with GA_3_ (*p* < 0.001; Figure 1). When planted in the fall in shallow rows, thickleaf penstemon had 12 times higher emergence when coated with GA_3_ ($\bar{x}$ = 1 plants m^−1^) than uncoated seeds ($\bar{x}$ = 0.08 plants m^−1^; *p* < 0.001). This increase in emergence with GA_3_ seed coating was more pronounced in deep furrows. Fall planting in deep furrows produced 29 times higher emergence from seeds coated with GA_3_ ($\bar{x}$ = 2.42 plants m^−1^) than uncoated seeds ($\bar{x}$ = 0.08 plants m^−1^; *p* < 0.001). Overall, 21 times more thickleaf penstemon emerged when planted in the fall ($\bar{x}$ = 0.90 plants m^−1^) than when planted in the spring ($\bar{x}$ = 0.04 plants m^−1^; *p* < 0.001). When planted in the spring, thickleaf penstemon only emerged when coated with GA_3_ and planted in shallow rows. However, overall, two times more thickleaf penstemon emerged when planted in deep furrows ($\bar{x}$ = 0.63 plants m^−1^) than in shallow rows ($\bar{x}$ = 0.31 plants m^−1^; *p* = 0.025; Figure 1).

The density of established thickleaf penstemon at the end of summer was also highest for GA_3_-coated seeds planted in the fall in deep furrows (Figure 2). Thickleaf penstemon had little to no establishment across all planting season and depth combinations unless coated with GA_3_ (*p* < 0.001; Figure 2). When planted in the fall in shallow rows, thickleaf penstemon had 12 times higher establishment when coated with GA_3_ ($\bar{x}$ = 1 plants m^−1^) than uncoated seeds ($\bar{x}$ = 0.08 plants m^−1^; *p* < 0.001). Additionally, no seedlings from deep furrows established unless the seed was coated with GA_3_ (*p* < 0.001). For fall planting, there was 27 times higher establishment from seeds coated with GA_3_ and planted in deep furrows ($\bar{x}$ = 2.25 plants m^−1^) than from uncoated seeds planted in shallow rows ($\bar{x}$ = 0.08 plants m^−1^; *p* < 0.001). Additionally, two times more thickleaf penstemon established overall when planted in deep furrows ($\bar{x}$ = 0.63 plants m^−1^) than in shallow rows ($\bar{x}$ = 0.31 plants m^−1^; *p* = 0.026; Figure 2).

### 2.3. Firecracker Penstemon

For firecracker penstemon, fall planting (*F* = 45.5, *p* < 0.001) and deep furrows (*F* = 22.1, *p* < 0.001) improved seedling emergence, but GA_3_ seed coating did not influence emergence (*p* = 0.92; Figure 1). The greatest seedling emergence for this species resulted from the combination of fall planting in deep furrows. There was a significant interaction between planting season and furrows (*F* = 3.8, *p* = 0.01). For spring planting, emergence was not different between shallow rows and deep furrows (*p* = 1), but for fall planting, deep furrows had greater emergence than shallow rows. For fall planting, four times more seedlings emerged from deep furrows ($\bar{x}$ = 3 plants m^−1^) than shallow rows ($\bar{x}$ = 0.83 plants m^−1^; *p* < 0.001; Figure 1).

Similar to seedling emergence, the establishment of firecracker penstemon at the end of the growing season was highest for seeds planted in the fall in deep furrows. There was a significant interaction between planting season and furrows (*p* < 0.001). For spring planting, establishment was not different between shallow rows and deep furrows (*p* = 1), but for fall planting, deep furrows had greater establishment than shallow rows. For fall planting, three times more plants established from deep furrows ($\bar{x}$ = 2.46 plants m^−1^) than from shallow rows ($\bar{x}$ = 0.71 plants m^−1^; *p* < 0.001; Figure 2). GA_3_ seed coating did not improve plant establishment over uncoated seeds for any planting season and furrow combination (*p* = 1; Figure 2).

## 3. Discussion

### 3.1. GA_3_ Seed Coating

Our prediction that GA_3_ seed coating would increase the emergence and establishment of Palmer’s, thickleaf, and firecracker penstemon was partially supported. In this study, GA_3_ increased the emergence and establishment of Palmer’s and thickleaf penstemon. These results build on other studies which show that the germination of these species increases when the seeds are treated with GA_3_ [11,17,36]. Although these studies also show that GA_3_ may increase the germination of firecracker penstemon [11,36]; in our study, GA_3_ coating did not improve the emergence or establishment of that species.

Differences in seed dormancy among the plant materials in this study may have influenced the response of GA_3_ seed coatings. Seed dormancy can be a complex, species-specific characteristic influenced by the environmental conditions where plant material was obtained [11,37,38]. Plant materials from higher elevations tend to have stronger seed dormancy [9,39,40]. Our firecracker seed was sourced from 1770 m, whereas our Palmer’s and thickleaf penstemon seeds were obtained from higher sites at 1890 m and 2130 m, respectively. As a heritable trait, seed dormancy may be influenced by the maternal environment where the seed was grown [41,42]. Penstemon seeds produced by cultivated plants tend to germinate more readily without treatment than penstemon collected from wild populations [37,39]. In our study, the firecracker seed lot was three generations removed from the wild collection, whereas both the Palmer’s and thickleaf penstemon seed lots were wild collections. The maternal environment of the seed production plot likely differed from that of the wild seed collection sites. This difference in the maternal environments may have influenced the inherited dormancy of the penstemon species used in this study [41,42]. Assuming the plant materials we tested in this study had differences in seed dormancy, we expect GA_3_ seed coatings to be most successful in the restoration of species with stronger seed dormancy.

Although we can decrease the dormancy of some species with GA_3_ coatings, it may not be in the best interest of restoration practitioners to break the dormancy of all seeds before planting. Evolutionarily, dormancy benefitted these species by allowing them to postpone germination to years when they would be more likely to establish [10,11,12,19]. A bet-hedging strategy may increase long-term restoration success by planting a combination of GA_3_-coated and uncoated seeds [10,43]. This would provide a mix of dormancy characteristics, so if the planting year is wet and favorable, the GA_3_-coated seeds can emerge and establish. If the planting year is unfavorable, however, the uncoated, dormant seeds may remain viable in the seedbank and germinate when conditions are better for plant survival [37].

### 3.2. Planting Season

Our prediction of increased emergence from spring planting of GA_3_-coated seed was not supported. Spring planting may have failed because of dry field conditions due to drought restrictions from January through June 2022 (Figure 3) [44]. Spring planting of GA_3_-coated seeds may be more successful in years with higher precipitation. However, high precipitation levels can make spring planting difficult due to muddy or frozen soils that make the site inaccessible to planting equipment [2,30]. Future studies are merited to test GA_3_-coated seeds in fall and spring plantings across multiple years to account for annual variations in moisture availability. Based on the findings of this study and the understanding that spring is a challenging period to plant, fall appears to be the best time to plant GA_3_-coated seeds.

### 3.3. Deep, U-Shaped Furrows

As predicted, deep furrows increased seedling emergence and plant establishment of all species of penstemon tested in this study. Presumably, increased seeding success from the deep furrows resulted from the treatment improving the microsite of the seed.

**Figure 3 plants-12-04005-f003:**
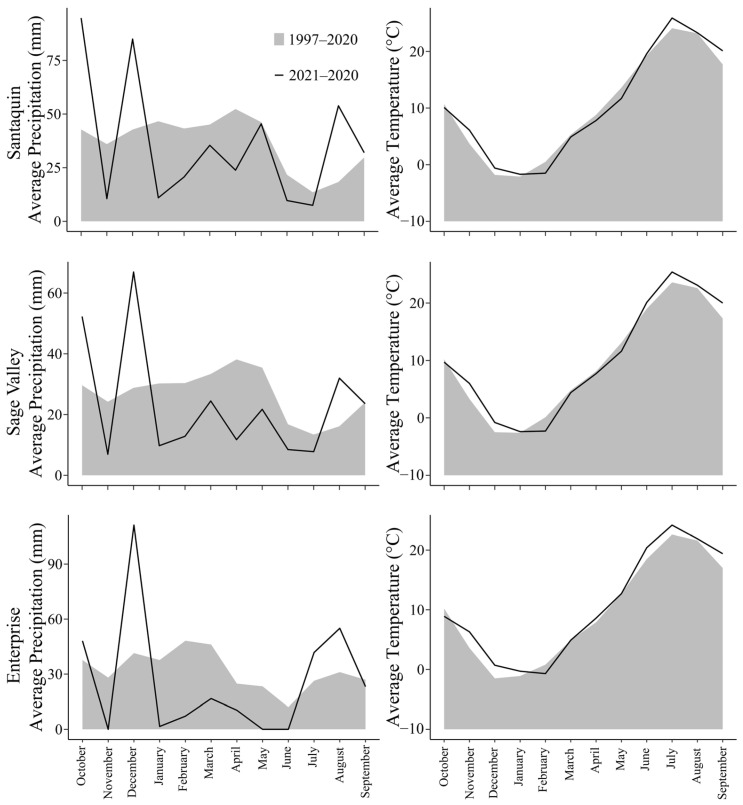
Precipitation and temperature data for Santaquin, Sage Valley, and Enterprise, Utah, USA from October 2021 through September 2022 compared to 30-year averages (1991 to 2020).

Anderson et al. measured increased soil moisture and moderated temperatures within deep, U-shaped furrows similar to the furrows in our study [27]. Optimal temperature and soil moisture contribute significantly to seed germination and seedling emergence [45]. Our results support the findings of other studies that showed increased emergence of grasses and Lewis flax (*Linum lewisii* Pursh) when planted in deep, U-shaped furrows [27,28]. Additionally, the U-shaped furrows were designed with a wider bottom than traditional V-shaped furrows to decrease the amount of soil that collapses onto the seed, which may bury seeds too deeply [27,28]. We did not see the decreases in penstemon emergence that we would expect if the sides of the furrows were collapsing and burying seeds too deeply. By creating a favorable microsite, the deep, U-shaped furrowing technique may improve the restoration success of small-seeded forbs, especially in areas where moisture is limited, and the re-establishment of native forbs is essential.

### 3.4. Future Work

Although these treatments substantially improved the emergence of penstemon, the percentage of seeds that emerged was quite low across our study. Low rates of emergence, or completely failed seedings, are common in native forb restoration in arid environments [2,3,22]. Despite the low emergence rates in our study, our statistically significant treatments had impacts that ranged from doubling emergence to increasing emergence by more than 20-fold. These increases to modest rates of emergence are significant steps in overcoming the many barriers to native forb restoration.

Additionally, low rates of emergence and establishment highlight the complexity of restoration. Seedling emergence was absent at Sage Valley and severely limited at Enterprise, UT, USA. Both Sage Valley and Enterprise are drier sites, and they received below-average precipitation from January through June 2022 (Figure 3) [44]. These sites may require years of higher precipitation to establish the penstemon species tested in this study. Thus, this research illustrates how the possibility of below-average precipitation should be considered when planning rangeland restoration projects [12,19,46]. Soil conditions at a restoration site may also be a concern [46]. At Sage Valley, thick soil crusts (>5 cm) developed, which likely prevented the emergence of small seedlings. Thus, when mineral soil crusts are present, additional treatments may be needed to restore an area [47].

The low establishment rate in our study may also be due, in part, to the small seed sizes of the species we tested. At our Santaquin site, where we had the greatest success, only 0.1% of all seeds emerged. In addition to low precipitation, the low emergence of our seeds may be partially explained by the tradeoff between energy investment per seed and the number of seeds produced [48]. Plants that have small seeds tend to yield greater quantities of seeds with less investment into individual seeds [48]. With this variation in parental investment, larger seeds correlate with improved germination and fitness, and smaller seeds tend to have lower success rates [48,49,50]. Like many native forb species, the penstemon seeds we planted were quite small (~1–3 mm). Successful restoration of disturbed, dry areas, such as our sites, will likely require a multifaceted approach that addresses environmental, soil, and seed characteristics [46].

With a broad range of factors limiting restoration success (e.g., dormancy, pathogens, drought, and soil crusting), compounding several treatments may be the most effective way to establish native forbs in disturbed and water-limited environments. Our study showed additive effects with GA_3_ coating, fall planting, and deep, U-shaped furrows. For Palmer’s and thickleaf penstemon, the highest emergence and establishment was consistently displayed by GA_3_-coated seeds planted in deep furrows in the fall. Combining seed treatments (e.g., GA_3_ seed coating) and planting techniques (e.g., deep, U-shaped furrows) to address multiple limiting factors may allow restoration practitioners to establish more native forbs in their restoration efforts.

## 4. Materials and Methods

### 4.1. Study Sites

This study occurred at three sites across Utah that had been altered by farming, grazing, or fire. From November through December 2021, studies were planted near Santaquin (39.9107, −111.8124), Sage Valley (39.5462, −112.0683), and Enterprise (37.5889, −113.7157), Utah, USA (Figure 4). These sites were planted again in March 2022. The Santaquin, Sage Valley and Enterprise sites were located at elevations of 1560, 1500, and 1620 m, respectively. The Santaquin site is classified as an upland stony loam, Sage Valley is an upland loam, and Enterprise is a semidesert shallow loam [51]. Additional site information is available in Table 1. From June to October 2021, we used glyphosate and 2,4-D to control existing vegetation prior to planting.

These sites receive limited precipitation every year. Historic annual precipitation for Santaquin, Sage Valley, and Enterprise was 438 mm, 321 mm, and 384 mm, respectively, based on 30-year averages [44]. Precipitation for 2021 to 2022 was below average for all sites (Figure 3). Santaquin, Sage Valley, and Enterprise received 430 mm, 278 mm, and 315 mm of precipitation, respectively, from October 2021 through 2022. While precipitation was lower than average for most months, precipitation was higher than average in October 2021, December 2021, and August 2022 (Figure 3). Mean temperatures during this study were similar to the 30-year averages for all sites (Figure 3).

After initial data collection, we excluded the data from Sage Valley and Enterprise, Utah, USA from analyses because emergence was severely limited or absent for most treatments at those sites. There was no emergence at Sage Valley, and only 12 of 144 rows (8%) contained seedlings at Enterprise. For these reasons, all results presented here come from the Santaquin site.

### 4.2. Seed Treatments

We obtained Palmer’s penstemon, thickleaf penstemon, and firecracker penstemon seeds from the Utah Division of Wildlife Resources Great Basin Research Center and Seed Warehouse in Ephraim, UT, USA. To evaluate the effectiveness of GA_3_ seed coatings, we compared seeds coated with a polymer imbibed with GA_3_ to uncoated seeds. The GA_3_ polymer was prepared by impregnating ethylcellulose (Ethocel™, Dow Chemical, Midland, MI, USA) with GA_3_ (Gold Biotechnology, St. Louis, MO, USA). To prepare this polymer, we dissolved 4.62 g of ethylcellulose in 50 mL of acetone on a stir plate. At the same time, we dissolved 0.382 g of GA_3_ in 10 mL of ethanol on a stir plate. We then mixed the ethylcellulose and GA_3_ solutions and stored the polymer in a sealed Erlenmeyer flask in a 4 °C cooler until we coated the seed. This rate was chosen after preliminary laboratory trials showed, in Petri dishes, that this application rate was sufficient to cause a germination treatment response.

We coated the seeds in 200 g batches using a 31 cm rotary seed coater (Universal Coating Systems, Independence, OR, USA) following standard seed coating protocols [52,53] (Halmer 2008, Pedrini et al. 2017). We used a 45% polyvinylpyrrolidone (PVP) solution (Agrimer-15, Ashland Inc., Covington, KY, USA) as a binder, prepared by mixing 45 g of PVP powder for every 100 g of water. First, we added 50 mL of GA_3_ solution directly to the atomizer disk by syringe and allowed the polymer to dry onto the seed. The acetone and ethanol from that solution evaporated in less than 20 s. Then, we added calcium carbonate powder (Clayton Calcium, Parma, ID, USA) over the seed while pumping the binder onto the atomizer disk. The amounts of lime and binder applied to each species varied by size and surface area of the seed (Table 2). Coated seed was then dried at 20 to 25 °C on a forced-air dryer (Universal Coating Systems, Independence, OR, USA) for approximately 10 min.

### 4.3. Field Planting

We planted our study in the fall (between 20 November and 8 December 2021) and spring (from 3 March to 22 March 2022). The study was a randomized split–split plot design with 6 blocks. Each block was randomly divided between planting in deep, U-shaped furrows (hereafter “deep furrows”) and shallow drill rows (hereafter “shallow rows”). These split plots were further divided into fall- and spring-planted plots. Species and treatments were randomized within split–split plots.

We planted the seeds using a JD Four-Row Planter (Kincaid Equipment Manufacturing Inc., Haven, KS, USA), which was adjusted to plant 2 m long rows 0.5 m apart. We fit modified furrowers in front of the planting disks to create furrowed rows that were 30 cm wide and 10 cm deep. The furrowers were modified by cutting off their bottom edge to make furrows with a flattened bottom. The furrowers were removed for shallow rows. Before planting, the seeder was calibrated to plant seeds ~2 to 5 mm below the soil surface, and the depth was periodically checked during planting to ensure uniform planting across blocks and furrow treatments. All species were planted at a rate of 246 pure live seed m^−1^. The furrows for spring planting were made in the fall. Then, we returned to each site in March and planted the seeds by hand to avoid complications associated with using heavy equipment on wet spring soils. We took care to plant at the same depth (~2 to 5 mm) as the seeder was calibrated to plant in the fall. Emerged seedlings at each site were counted from 20 May to 3 June 2022. Established seedlings were counted on 10 August 2022.

### 4.4. Statistical Analysis

For the field study, we analyzed the proportion of seeds that emerged and the proportion of seeds that were established using generalized linear mixed effects models with a Poisson distribution. We constructed models for each species using the ‘glmer’ function of the ‘lme4′ package [54] in R [55]. Block was defined as a random effect. Treatment, furrows, and season of planting were defined as fixed effects. We tested for two-way interactions between treatment, season, and furrows for each model. Pairwise comparisons were then conducted using the Tukey method with the ‘emmeans’ function of the ‘emmeans’ package [56].

## 5. Patents

Larson, A.J.; Michaelis, D.; Madsen, M.D., 2022. Development and Use of a Slow-Release Polymer Seed Coating System to Deliver Growth Hormones for Enhancing Seed Germination and Early Plant Growth. Provisional Patent Application No. 63317605.

## Figures and Tables

**Figure 1 plants-12-04005-f001:**
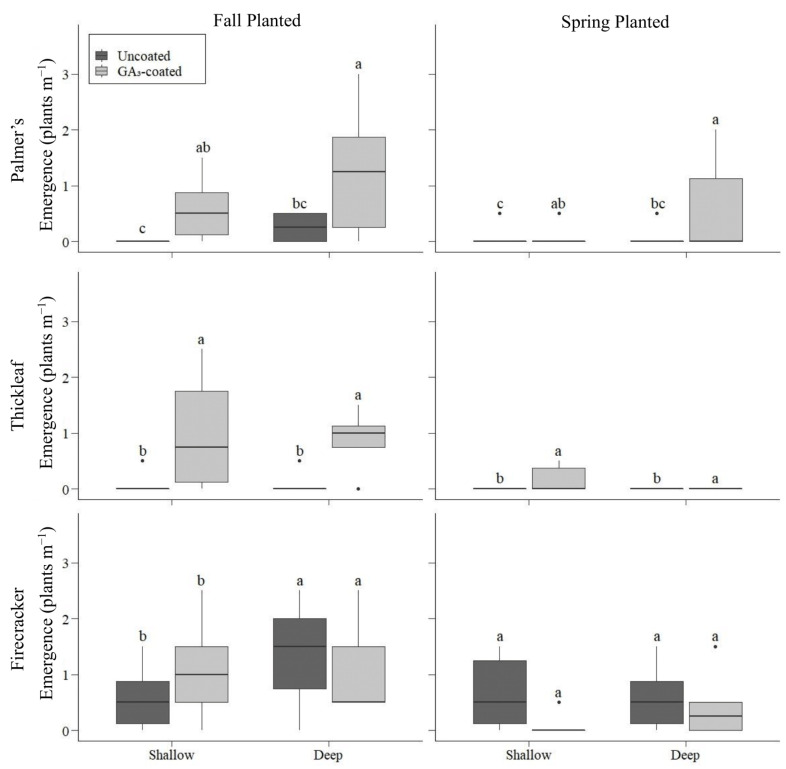
Emergence (plants m^−1^) of penstemon species at Santaquin, Utah, USA by season of planting, planting depth, and seed treatment. Thick horizontal lines within boxes represent medians. Dots represent values >1.5 times the interquartile range above the 75th percentile or below the 25th percentile. Differences at the *p* < 0.05 level are indicated by different letters within species and seasons.

**Figure 2 plants-12-04005-f002:**
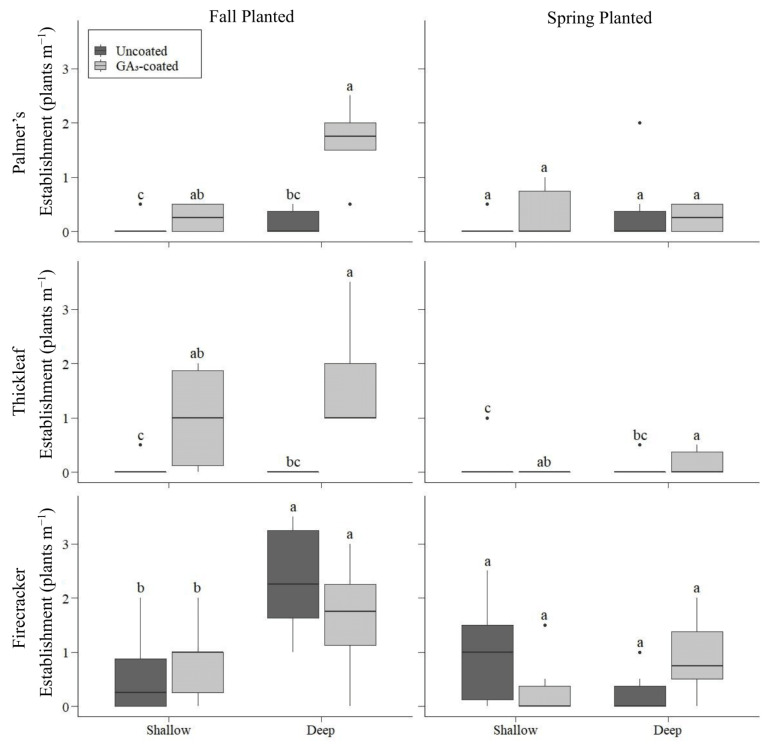
Establishment (plants m^−1^) of penstemon species at Santaquin, Utah, USA by season of planting, planting depth, and seed treatment. Thick horizontal lines within boxes represent medians. Dots represent values >1.5 times the interquartile range above the 75th percentile or below the 25th percentile. Differences at the *p* < 0.05 level are indicated by different letters within species and seasons.

**Figure 4 plants-12-04005-f004:**
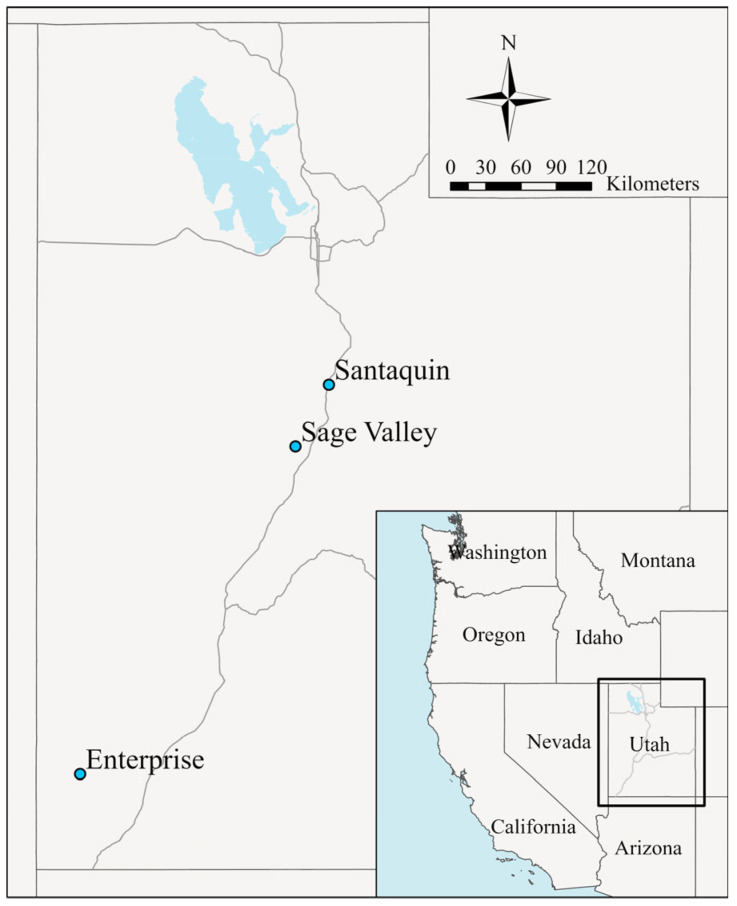
Locations of planting sites at Santaquin, Sage Valley, and Enterprise, UT, USA.

**Table 1 plants-12-04005-t001:** Descriptions of soil, temperature, and precipitation from Santaquin, Sage Valley, and Enterprise, Utah, USA. Climate data are based on 30-year averages from PRISM models [44]. Soil data are sourced from the web soil survey [51].

Site	Soil Map Unit	Mean Annual Precipitation (mm)	Low Mean Monthly Temperature (°C)	High Mean Monthly Temperature (°C)	Elevation (m)
Santaquin	Donnardo Stony Loam	475	−2	23.8	1560
Sage Valley	Juab Loam	349	−2.9	23.3	1500
Enterprise	Checkett-rock Outcrop Complex	362	−1.4	22.6	1620

**Table 2 plants-12-04005-t002:** The amount and type of coating materials used to apply gibberellic acid coating for the penstemon species used in this study.

Penstemon Species	45% PVP *	Calcium Carbonate	Gibberellic Acid	Ethylcellulose	Acetone	Ethanol
	---------------------------------g--------------------------------				----------mL--------	
Palmer’s	47	200	0.382	4.62	50	10
Thickleaf	52	300	0.382	4.62	50	10
Firecracker	58	200	0.382	4.62	50	10

* PVP = polyvinylpyrrolidone.

## Data Availability

The data presented in this study are openly available at Brigham Young University ScholarsArchive at https://scholarsarchive.byu.edu/data/55, accessed on 4 November 2023.

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
