# Peer review of "Breaking Dormancy and Increasing Restoration Success of Native Penstemon Species Using Gibberellic Acid Seed Coatings and U-Shaped Furrows"

_plants, 2023, doi:10.3390/plants12234005_

Round 1

Reviewer 1 Report

Comments and Suggestions for Authors

Title - I suggest that the title should include Penstemon species to be more accurate.

 Abstract - why do the first two sentences refer to some plants and not specifically to Penstemon species?

L.10 Did you want to emphasize that GA breaks dormancy and also affects germination? The possibility of germination is a consequence of the end of dormancy.

 Introduction - My feeling is that the first part of the introduction without quoting Penstemon species is too long, and too little attention is paid to Penstemon species, only 10 lines. I should write more about the plant e.g. about its importance, about the use of methods of breaking seed dormancy (e.g. stratification, other chemicals), about germination and seedling emergence. (e.g. Laufman J.E and Wisner1997; Kitchen p., Meyer 1991).

 L.86 emergence and establishment seeds - is this the right formulation? L.87, it seems to me that seed planted was used instead of sowing seeds.

 Results-Fig1 Emergence of species? -this is a simplification.

 Fig.2 Establishment .. species - this is a simplification. Chapters Results, Discussion and Future work – must be shortened.  Instead of Future it is better to introduce a Conclusions section.

Extremely low values were obtained for seedling emergence and seedling establishment. Do such results have any significance although are statisticaly significant, and can the technique be recommended to practitioners?

Material and Methods

The species used in the work are not specified.

It is not clear - how long were the seeds dried? Were they dried before and after treatment? Was the effect of acetone and alcohol on seed germination/seedling emergence checked? Concentration of GA in M should be added. It is not clear what constituted a repetition and how many repetitions there were. How many seeds were sown per meter? L.332-​ L. 332-Did you add solution containing GA, Polymer and acetone?

References-number 17  required  more  information

Author Response

Thank you for your time and effort in reviewing our manuscript and for your helpful feedback.

Title - I suggest that the title should include Penstemon species to be more accurate.

  • The title was adjusted to specify Penstemon species rather than just native forbs.

Abstract - why do the first two sentences refer to some plants and not specifically to Penstemon species?

  • Our aim was to focus on the technologies and techniques rather than to focus on the species of penstemon. The Penstemon species are important because we used them to test these technologies, but the big idea of this paper is the potential of this technology to help restore dormant species- not just penstemons. Everything talked about in the first two sentences applies to penstemons, but also to a broader picture and a broader audience.

L.10 Did you want to emphasize that GA breaks dormancy and also affects germination? The possibility of germination is a consequence of the end of dormancy.

  • Line 10 is worded to include both overcoming dormancy and increasing germination to specify to readers who may not be familiar with gibberellic acid that gibberellic acid increases germination by overcoming dormancy.

Introduction - My feeling is that the first part of the introduction without quoting Penstemon species is too long, and too little attention is paid to Penstemon species, only 10 lines. I should write more about the plant e.g. about its importance, about the use of methods of breaking seed dormancy (e.g. stratification, other chemicals), about germination and seedling emergence. (e.g. Laufman J.E and Wisner1997; Kitchen p., Meyer 1991).

  • As with the abstract, our aim was to focus on the technologies and techniques rather than to focus on the species of penstemon. The Penstemon species are important because we used them to test these technologies, but the big idea of this paper is the potential of this technology to help restore dormant species- not just penstemons. Everything talked about in the first two sentences applies to penstemons, but also to a broader picture and a broader audience.

L.86 emergence and establishment seeds - is this the right formulation? L.87, it seems to me that seed planted was used instead of sowing seeds.

  • We changed “planted” to “sown” (lines 87, 88, and 91).

Results

Fig.1 Emergence of species? -this is a simplification.

  • We added the units of emergence (plants m-1) to figure 1.

Fig.2 Establishment .. species - this is a simplification.

  • We added the units of emergence (plants m-1 ) to figure 2.

Chapters Results, Discussion and Future work – must be shortened.

  • This is a good point. Based on the relatively complex design of our study, however, we need to keep the material in these sections to explain all the relevant results for our different variables and key applications of those results.

Instead of Future it is better to introduce a Conclusions section.

  • This is a good point, which we have considered. The journal instructions indicate that the conclusions section is optional. We feel that the future work section is very meaningful as this research is only one stage in developing this technology, and reviewer 2 specifically commented on appreciating that section. Thus we concluded this manuscript by leading into what comes next in future work rather than a traditional conclusions section.

Extremely low values were obtained for seedling emergence and seedling establishment. Do such results have any significance although are statisticaly significant, and can the technique be recommended to practitioners?

  • This is an important point. We added lines 249-254 to address this question. As we explain, emergence and establishment values in our study are quite low. In addition to the statistical significance, we examined the magnitude of the effects. Native forb restoration success is very low (and often fails) in arid environments, so being able to double any emergence or increase emergence by 20-fold is significant for restoration practitioners. This could mean the difference between being unable to use forbs because none will establish and being able to get a few established on the landscape to start to repopulate an area. Additionally, as we discuss in lines 276-283, finding multiple treatments and techniques to address barriers to restoration may allow us to improve restoration success little by little.

Material and Methods

The species used in the work are not specified.

  • The species used are first introduced in the introduction (lines 79-81) along with the purpose for choosing those species. The species are again specified in the methods on line 321.

It is not clear - how long were the seeds dried? Were they dried before and after treatment?

  • We added that the seeds were dried for approximately 10 minutes at line 344.

Was the effect of acetone and alcohol on seed germination/seedling emergence checked? 

  • We added a sentence explaining how the acetone and ethanol evaporate off during the coating process (lines 338-339). The potential treatment effects of acetone and alcohol is an interesting point. This interaction is outside the scope of our field evaluation but is a good idea for future laboratory research.

Concentration of GA in M should be added. 

  • The GA we used was in a powder form and we provide the mass measurements of GA we used to create our polymer solution (lines 328). All the measurements needed for others to replicate our GA solution are provided in lines 327-328.

It is not clear what constituted a repetition and how many repetitions there were. 

  • We added the number of blocks to line 351. The blocks are then defined in lines following.

How many seeds were sown per meter? 

  • As stated in line 362, all species were planted at a rate of 246 pure live seeds m-1.

L.332-Did you add solution containing GA, Polymer and acetone?

  • In lines 326 – 330, we describe the creation of the solution by mixing ethylcellulose (a polymer), acetone, GA, and ethanol. Line 337 explains how we added 50mL of that solution to the seeds.

​References-number 17  required  more  information

  • Reference 17 was updated to include the information “Provisional Patent Application No. 63317605” (line 429-430).

Reviewer 2 Report

Comments and Suggestions for Authors

The Authors submitted to Plants an article about regarding possible strategy for breaking dormancy and increasing restoration of native forbs, Penstemon, by Gibberellic acid coatings and U-shaped furrows
The topic of the article is very interesting and timely, because it addresses a problem extremely connected to the maintenance of ecosystems.
In the paper the Authors introduce the topic of their study well with an adequate and updated bibliography. The results are clearly represented graphically and are well described and discussed, leading to shareable conclusions.
I appreciate that they dedicate an entire paragraph of the discussion to present future work resulting from this study.
Only two questions:
-    How was the quantity of GA3 to produce this coating determined, why that one? In my opinion, some explanation regarding this should be included in the paper
-    Do you plan to evaluate some biological/biochemical parameters to analyse at a molecular level the breaking dormancy effect of the strategies you have tested??

Author Response

Thank you for time and effort reviewing our manuscript.

How was the quantity of GA3 to produce this coating determined, why that one? In my opinion, some explanation regarding this should be included in the paper

  • We added information that preliminary laboratory trials had shown that this rate was sufficient to cause a germination treatment response (lines 330-332).

Do you plan to evaluate some biological/biochemical parameters to analyse at a molecular level the breaking dormancy effect of the strategies you have tested??

  • We did not evaluate those parameters at a molecular level as this was outside the scope of our study, but we think this would be an interesting idea for future studies.

Reviewer 3 Report

Comments and Suggestions for Authors

The manuscript presents an interesting, multistep study of the modulation of strong seed dormancy. The study contributes to the search for an effective method to apply hormone coating to the seeds outside the laboratory, as well as the creation of an appropriate microsite for efficient plant germination. As the results could be implemented in the field to restore native forbs in dry, disturbed areas, the topic of the research is valid. In my opinion, this paper would be interesting not only as the basis for further research in the field area of the Authors but also for the other research groups. 

Generally, the study is well designed and interpreted, can arouse public interest, and can be considered for publication. I read the manuscript thoroughly and I could not find any serious allegations therefore I would recommend it for publication in Plants journal. 

Author Response

Thank you for your time reading and reviewing our manuscript.

Reviewer 4 Report

Comments and Suggestions for Authors

Manuscript entitled "Breaking Dormancy and Increasing Restoration Success of Native Forbs Using Gibberellic Acid Seed Coatings and U-shaped Furrows" submitted to Plants journal is well written. The paper is adequately organized and the topic is interesting and focuses on techniques restoration after seed dormancy, so could by used  to restore native forbs in dry, disturbed areas. Although the manuscript is well-edited, however improvements should be introduced that will improve its quality:

- The title of the manuscript is too general - the study only covers three species of penstemon, not the larger population of native forbs

- In my opinion chapter 1. Introduction is too long - should be cuted to just only major information

- Tab. 1. Two columns have the same caption: "Low Mean Monthly Temperature" this should be corrected

- Figure 2. The units should be uniform throughout the manuscript - please change the notation plants/m to the SI unit as it is in the text i.e. plants m-1

- The manuscript is missing chapter 6. Conclusions - this should be completed

Author Response

Thank you for your time reviewing our manuscript and especially for your valuable feedback and suggestions.

The title of the manuscript is too general - the study only covers three species of penstemon, not the larger population of native forbs

  • The title was adjusted to specify Penstemon species rather than just native forbs.

In my opinion chapter 1. Introduction is too long - should be cuted to just only major information

  • Due to the relatively complex nature of our study design, we needed to include several paragraphs addressing different factors incorporated into our study. In the first paragraph we explain a problem in restoration that we are trying to alleviate. The second paragraph addresses our use of GA seed coatings, the third paragraph addresses our testing of multiple planting seasons, and the fourth paragraph addresses our use of deep, U-shaped furrows. The fifth paragraph introduces our test species and why they were chosen, and the sixth explains our predictions.

Table 1. Two columns have the same caption: "Low Mean Monthly Temperature" this should be corrected

  • The column of the table was corrected to say “High Mean Monthly Temperature” (page 8).

Figure 2. The units should be uniform throughout the manuscript - please change the notation plants/m to the SI unit as it is in the text i.e. plants m-1

  • These figures were updated to use plants m-1 as the units (pages 3-4).

The manuscript is missing chapter 6. Conclusions - this should be completed

  • This is a good point, which we have considered. The journal instructions indicate that the conclusions section is optional. We feel that the future work section is very meaningful as this research is only one stage in developing this technology, and reviewer 2 specifically commented on appreciating that section. Thus we concluded this manuscript by leading into what comes next in future work rather than a traditional conclusions section.

Round 2

Reviewer 1 Report

Comments and Suggestions for Authors

Thank you very  much  for using remarks.

1."we dissolved 0.382 g of GA3 in 10 mL of ethanol" I suggest adding the molar concentration in brackets.
2." First, we added 50 mL of GA3 solution" -add molar concentration

3. Are you sure the alcohol evaporated within 20 seconds?